# Three-Year Follow-Up Detecting Choroidal Neovascularization with Swept Source Optical Coherence Tomography Angiography (SS-OCTA) after Successful Half-Fluence Photodynamic Therapy for Chronic Central Serous Chorioretinopathy

**DOI:** 10.3390/diagnostics13172792

**Published:** 2023-08-29

**Authors:** Olivia Esteban-Floría, Guillermo Pérez-Rivasés, Ana Honrubia-Grijalbo, Isabel Bartolomé-Sesé, María Dolores Díaz-Barreda, Ana Boned-Murillo, Pablo Cisneros-Arias, Javier Mateo-Gabás, Francisco-Javier Ascaso-Puyelo

**Affiliations:** 1Department of Ophthalmology, Hospital Clínico Universitario Lozano Blesa, 50009 Zaragoza, Spain; oliviaestebanfloria@hotmail.com (O.E.-F.); anahonrubia@telefonica.net (A.H.-G.); issbartolome@gmail.com (I.B.-S.); lodiba92@gmail.com (M.D.D.-B.); anabomu@hotmail.com (A.B.-M.); pablo_cisneros14@hotmail.com (P.C.-A.); jmateo_1999@yahoo.com (J.M.-G.); jascaso@gmail.com (F.-J.A.-P.); 2Aragon Health Research Institute (IIS Aragón), 50009 Zaragoza, Spain

**Keywords:** swept-source optical coherence tomography angiography (SS-OCTA), neovascular chronic central serous chorioretinopathy, photodynamic therapy

## Abstract

**Purpose**: To assess the clinical course, structural changes, and choroidal neovascularization detection by SS-OCTA in long-standing and resolved patients with chronic central serous chorioretinopathy (cCSC) after successful half-fluence photodynamic therapy (hf-PDT) treatment. **Methods:** Twenty-four eyes presenting with cCSC were examined with SS-OCTA and were classified as choroidal neovascular (CNV) or non-choroidal neovascular (non-CNV) cCSC depending on the vascular pattern detected by SS-OCTA after one, two, and three years after hf-PDT. Two groups were compared based on the following clinical findings: demographic characteristics, time of clinical signs, best corrected visual acuity (BCVA), central retinal thickness (CRT), central choroidal thickness (CFT), subretinal fluid (SRF), flat, irregular pigment epithelial detachment (FIPED), and features of fluorescein angiography (FA) and vascular pattern by SS-OCTA. **Results:** All patients showed resolved cCSC during follow-up after hf-PDT. A total of 5 of 24 (20.8%) eyes showed a neovascular pattern by SS-OCTA. No differences between BCVA, CRT, SRF, FIPED, or FA features were found between both groups (*p* > 0.05). However, CFT and older age were associated with a neovascular pattern by SS-OCTA (*p* < 0.05) in follow-up. No signs of neovascular activity were detected by SS-OCT during follow-up in CNV cCSC patients, and no antiVEGF treatment was required for three-year follow-ups. **Conclusions:** Despite patients with cCSC showing a favorable clinical response after hf-PDT, lower foveal thickness and older age were associated with CNV patterns by SS-OCTA during follow-up.

## 1. Introduction

Chronic central serous chorioretinopathy (cCSC) is a common cause of vision loss in young adults characterized by an accumulation of subretinal fluid (SRF), atrophic changes to the outer retina or retinal pigment epithelium (RPE), and is frequently associated with flat, irregular pigment epithelial detachment (FIPED) mainly in the macular area due to choroidal vascular hyperpermeability [1].

Photodynamic therapy (PDT) is considered first-line after less invasive methods (corticosteroids, eplerenone) have not been effective [2] with a high percentage (85–100%) of SRF resolution and no serious adverse events; additionally, the subthreshold laser can be used for this matter. However, CNV was described after laser photocoagulation or photodynamic therapy (PDT) for CSC [3]. The mechanisms underlying secondary choroidal neovascularization (CNV) in cCSC remain unclear. It has been considered a complication of the evolution of this disease with a challenge for its diagnosis with a variable incidence depending on diagnostic criteria [4,5,6,7]. Recently, the visualization of FIPED by OCT has been associated with CNV by OCT-A in cCSC in different studies. The overlap of imaging features between CNV and cCSC manifestations often challenges the detection by OCT, fluorescein angiography, or indocyanine green angiography [7,8]. The use of swept-source OCT angiography as a non-invasive imaging technology that enables visualization of three-dimensional retinal and choroidal vasculature has increased the incidence of CNV detection rate in cCSC [8].

In this study, we included cCSC eyes successfully treated with hf-PDT and followed up for three years to report clinical course, structural changes, and incidence of CNV to further evaluate factors associated with vascularized cCSC patients.

## 2. Materials and Methods

### 2.1. Patients

This prospective case series included 24 naïve patients with cCSC (lasting longer than 6 months after the onset of the first symptom) who received hf-PDT therapy for foveal subretinal serous detachment between January 2016 and 2019. The study was conducted at the Ophthalmology Department in University Clinic Hospital Lozano Blesa, Zaragoza (Spain).

The diagnosis was based on the patient’s symptoms and clinical signs of cCSC longer than 6 months, visual acuity, fundoscopic examination, fluorescein angiography, indocyanine green angiography, and swept-source-OCT (SS-OCT) and SS-OCT angiography (SS-OCTA) (DRI OCT Triton, Topcon, Tokyo, Japan).

Subjects were excluded if their symptoms lasted fewer than six months; exhibited any sign of CNV with fluorescein angiography, indocyanine green angiography, or OCTA; bad response to hf-PDT (SRF at 6-month visit), focal laser photocoagulation, or anti-VEGF treatment; had evidence of choroidal atrophy; had high myopia defined as a refractive error (spherical equivalent) < −4.0 diopters or an axial length > 24.5 mm; underwent continuous corticosteroid therapy; or had intraocular surgeries, poor signal strength, or severe artifacts due to saccadic eye movement.

A total of 24 eyes of 24 patients received hf-PDT in one eye, and they have been followed up for a minimum of three years. Each patient underwent a comprehensive ophthalmological examination, including a review of medical and clinical history, measurement of best-corrected visual acuity (BCVA), slit-lamp biomicroscopy, intraocular pressure using Goldman applanation tonometry, and refraction assessment. BCVA was measured using a standard retro-illuminated Early Treatment Diabetic Retinopathy Study (ETDRS) chart. These clinical parameters were obtained from 12, 24, and 36-month visits follow-up after hf-PDT.

The research protocol adhered to the Declaration of Helsinki, and the study obtained the approbation of the local ethics committee (Clinical Research Ethics Committee of Lozano Blesa University Clinic Hospital and the Clinical Research Ethics Committee of Aragon (CEICA). Informed consent was obtained from all studied subjects.

### 2.2. Half-Fluence Photodynamic Therapy

Hf-PDT was performed using half of the energy level used in the treatment of age-related macular degeneration (AMD) with protocol; 6 mg/m^2^ body surface area of verteporfin (Visudyne, Novartis AG, Basel, Switzerland) was injected intravenously over a period of 10 min; then, 15 min after the start of the injection, the lesion was irradiated with a laser for 83 s at an output of using a diode laser (Visulas 690 S; Carl Zeiss Meditec Inc., Dublin, CA, USA) capable of delivery at 689 nm light and a half fluence of 25 J/cm^2^ instead of 50 J/cm^2^. The area of irradiation included the area of choroidal hyperfluorescence on ICGA corresponding to the leaking points and RPE detachment on FA.

### 2.3. Image Acquisition

SS-OCT and SS-OCTA were performed using swept-source OCT with deep range imaging (DRI)-Triton SS-OCT (Topcon Corporation, Tokyo, Japan). Retinal and choroidal thickness maps were overlapped to the ETDRS grid (6 × 6 mm) to obtain values for each sector. Builtin software was used to automatically calculate retinal and choroidal thickness values in the ETDRS grid; the inner and outer rings, with semidiameters of 1500 μm and 3000 μm, respectively, were segmented into four quadrants (superior, inferior, nasal, and temporal). The central sector was defined as being within 1000 μm of the foveal center. SS-OCTA was performed with 3 × 3 mm protocol IMAGEnet 6 Version software 1.22.1.14101^®^ by OCT angiography (Topcon Corporation, Tokyo, Japan).

### 2.4. Data Collected

The data collected included patient gender, age (years) at diagnosis, BCVA, presence or absence of subretinal fluid (SRF), flat pigment epithelium detachment (DEPs), and retinal or choroidal thickness (CRT or CFT) at 12, 24, and 36-month visit after successful treatment with hf-PDT.

### 2.5. Statistical Analyses

Statistical Product and Service Solution (SPSS; v21, IBM Corp. Armonk, NY, USA) was used for statistical analysis, and p-values less than 0.05 were considered statistically significant.

The normal distributions of the characteristics of patients were assessed using the Kolmogorov–Smirnov test.

When comparing two groups, the chi-square test was used depending on the number of samples and whether the assumption of a normal distribution was met. A Wilcoxon signed-rank test for paired data was used to compare pre- and post-hf-PDT results.

## 3. Results

From February 2017 to July 2019, from 30 patients with cCSC undergoing regular follow-up at our clinic, a total of 24 eyes from 24 patients (male: female = 18: 6) fulfilling the inclusion criteria were enrolled in this study. The mean age at diagnosis was 47.42 ± 9.02 years. The mean duration of symptoms was 22.08 ± 8.20 months before treatment. Of the 24 eyes, 23 had no SRF during follow-up, and 1/24 had SRF at the 36-month visit. All patients received one session of hf-PDT.

During follow-up, we established two groups: CNV CSC and non-CNV CSC, according to SS-OCTA during follow-up. In 5 of the 24 eyes (20.8%), an outer retinal SS-OCT angiogram revealed the presence of an abnormal vessel configuration, showing a distinct CNV network under retinal pigment epithelium (RPE), suggesting the presence of secondary CNV. A qualitative comparison of the SS-OCTA images with the SS-OCT B-scan revealed that the neovascular networks corresponded to a flat hyperreflective pigment epithelial detachment (FIPED) in all patients with neovascular networks (Figure 1).

The mean time between the initial presentation of CNV by SS-OCTA was 4/5 in the 12-month visit, and in 1/5 of patients, it was detected in a 36-month visit by SS-OCTA.

In 4/5 patients, no signals of activity (subretinal or intraretinal fluid) by SS-OCT were shown, and no antiVEGF treatment was required. Only one of the patients showed SRF at the last visit at 36 months.

Differences in age, duration of symptoms and features of FA and presence or absence of FIPED in SS-OCT are shown in Table 1. The mean older age of the CNV CSC group was statistically significant in relation to the non-CNV CSC group (*p* < 0.05). We found worse mean BCVA in the CNV CSC group than in the non-CNV CSC group, but these differences were not statistically significant; Table 2. There were no differences between groups in the duration of disease until hf-PDT, features in FA, or presence or absence of FIPED in SS-OCT (*p* > 0.05). No differences were found between both groups in CRT; however, thinner foveal choroidal thickness was found in CNV CSC patients (*p* < 0.05) in all visits; Table 3.

## 4. Discussion

Chronic CSC is a sight-threatening disease that may be complicated by secondary CNV, representing a diagnostic challenge with conventional imaging modalities because CNV signs can overlap with those of chronic CSC [9,10]. We used SS-OCTA in the attempt to identify CNV in resolved cCSC after hf-PDT. We also use clinical and SS-OCT features about the retina and choroid layers to identify differences between CNV CSC and non-CNV CSC patients. SS-OCT and SS-OCTA have higher wavelengths and scan rates, resulting in a higher definition of images and better contrast of small neovascular structures [11]. Several studies have demonstrated that OCTA increases the detection of CNV in chronic CSC compared to conventional diagnostic modalities [3,5,12,13,14].

We identified in five patients a flat and hyperreflective PED by SS-OCT, which coexists directly with CNV patterns by SS-OCTA. Some authors have already associated it before with choroidal neovascularization (CNV) in cCSC. The incidence ranges from 2.4% to 58.0%, depending on the authors and the process of diagnosis [4,5,9,15,16,17]. Our results are similar to these studies, with an incidence of 24.5% by SS-OCTA. In our study, none of them presented signs of activity, and no anti-VEGF treatment was required in follow-up. Only one of them had SRF in a 36-month visit, and it was treated with a second hf-PDT after finishing the follow-up.

Like our results, some authors described a low-grade CNV activity secondary to CSC [16], and quiescent CNV from pachychoroid neovasculopathy was previously reported. Querques and colleagues found one-third of vascularized FIPED areas were symptom-free and fluid-free, which could be called quiescent CNV [18]. Chen and colleagues reported that the clinical course of CNV in CSC was stable during a three-year follow-up after PDT treatment. Guo. and colleagues demonstrated that the FIPED-associated CNV remained stable without SRF recurrence in the follow-up after half dose-PDT [19]. Furthermore, they concluded that vascular FIDED is supposed to be quiescent in patients with chronic CSC treated with half-dose PDT.

We collected the data of BCVA during follow-up. In our case, we described differences in mean BCVA between groups in basal measure. However, these differences were not statistically significant. During follow-up, we also showed the evolution of the BCVA in both groups. We observed a stable vision in both groups with no significant differences between visits for 36 months. There are few reports in the literature regarding the long-term prognosis in patients with CSC complicated by CNV. It has been reported that CSC was associated with poor long-term visual prognosis [20,21] and that the presence of CNV was associated with lower vision when comparing data of patients with and without CNV [22]. However, according to our results, Kim et al. recently observed that there was no statistically significant change in the BCVA after a 5-year follow-up, and the patients with CNV detection demonstrated stable and good vision during the long-term [23].

Pachychoroid neovasculopathy was described as a disease characterized by a form of type 1 neovascularization involving dilated choroidal vessels in areas of increased choroidal thickness. The authors support their hypothesis that eyes with long-standing “silent” pachychoroid disease (because it is asymptomatic) may develop type 1 neovascularization in the absence of CSC manifestation. In our series, all patients were good responders to hf-PDT, and no SRF was found in follow-up, so they may be considered as patients with silent pachychoroid disease with a similar association [15,24,25].

There are different hypotheses to describe the pathogenesis of secondary CNV in cCSC. Anatomical and physiological changes originating from CSC (choroidal hyperperfusion, vascular dilation, and succedent RPE abnormalities) increase levels of vascular endotelial growth factor (VEGF) in the sub-RPE compartment and share genetic susceptibility [21]. Other authors affirm a compensatory role of CNV in chronic CSC: the long-term ischemic outer retina that results from SRF (subretinal fluid) may be compensated by the growth of CNV (providing oxygen and nutrients to the outer retina) [26,27]. Photodynamic therapy (PDT) treatment has also been associated with secondary CNV in the literature because of its pro-inflammatory effect or reduction in chorioretinal perfusion [28,29]. Thus, various “safety-enhanced” PDT protocols have been devised to optimize treatment outcomes. They typically use reduced dose or reduced fluence. However, a proportion of patients develop secondary CNV even after half-dose PDT or after half-fluence PDT, as in our study. A decrease in choroidal perfusion after PDT may increase the risk of CNV development by promoting the release of VEGF [30].

In our study, we have shown that only some patients with resolved cCSC after hf-PDT have developed CNV in follow-up. Thus, other factors are promoting CNV in patients with favorable results after PDT. We studied if characteristics in FA before treatment may have an influence on developing CNV, but we have not found any significant differences.

Pigment epithelial detachment (PED) is frequently involved in the macular area in acute CSC [31,32], and flat, irregular pigment epithelial detachment (FIPED) has been associated with cCSC. Recently, FIPED has also been associated with CNV-cCSC by OCT-A [7,33]. We studied the number of patients with basal avascular FIPED by SS-OCT and SS-OCTA before PDT and follow-up for 36 months. The incidence of FIPED was 18 out of 24 patients, and we did not find significant differences in the number of basal FIPED between groups. However, 5 out of 24 developed CNV by SS-OCT-A. So, we showed how FIPED is not always related to the development of CNV, as other authors described before. Bousquet et al. reported that one-third of FIPED in CSC patients involved CNV on OCTA, and Faghihu et al. described vascular FIPED in 14 of 42 cCSC patients [7,34]. So, we hypothesize that other factors are involved in developing a vascular FIPED.

Choroidal thickness may also play an important role in CNV CSC [24]. Choroidal thickness after PDT has been found to be thinner in the CNV CSC group [8,18,34,35] than in the non-CNV CSC group. We agree with these studies; in ours, we showed that the CNV CSC group has thinner choroidal thickness than the non-CNV CSC group in 12, 24, and 36-month visits follow-up (*p* < 0.05).

Aging of the RPE–Bruch’s membrane complex as in AMD and longer history of chronic CSC, leading to chronic alterations of Bruch’s membrane and RPE, so vulnerability to CNV formation [36]. Other authors have already mentioned age as a risk factor for CNV in CSC [8,36,37,38]. In our study, mean age was significantly greater in CNV CSC patients than in non-CNV CSC patients.

The present study’s limitations are the limited number of patients because of long follow-up. However, all scanned patients’ high-quality SS-OCTA images could be generated, and the neovascular component could be clearly identified in the outer retina by four independent readers without manual adjustment of the segmentation lines in OCT B-scans. This highlights the SS-OCTA device’s practical clinical application for CSC patients. It should be noted that age can be considered a confounding factor, as it may be accompanied by choroidal thinning and, therefore, a consequent neovascular pattern. For this reason, during the disease, the spontaneous appearance of CNV in non-neovascular patterns could be associated without age as a risk factor, although it is already described as a risk factor by other authors, as we have mentioned above [8].

In conclusion, we showed that although patients with chronic CSC showed a favorable clinical response to hf-PDT, lower foveal thickness and older age can be associated with a CNV pattern by SS OCT-A in follow-up.

## Figures and Tables

**Figure 1 diagnostics-13-02792-f001:**
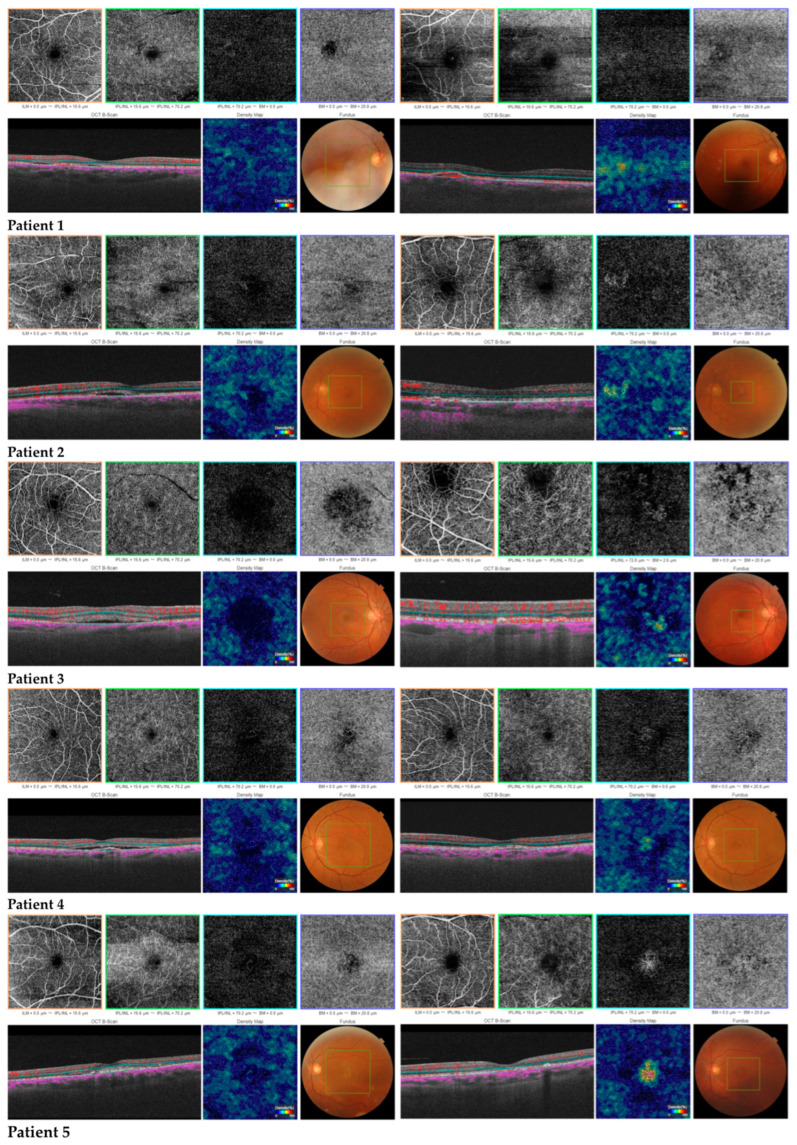
Follow-up of the neovascular pattern by SS-OCTA after hf-PDT in five patients.

**Table 1 diagnostics-13-02792-t001:** Basal demographic and clinical characteristics in both groups: NVC CSC and non-NVC CSC patients.

Diagnosis	CNV CSC (*N* = 5)	Non-CNV CSC (*N* = 19)	*p* ^1^
Age, years	51.6 (3.4) [47.3; 55.9]	46.3 (9.8) [41.6; 51.0]	0.044 *
	51.0 (7)	45.0 (10)	
Duration of symptoms, months.	21.6 (28.3) [−13.6; 56.8]	22.2 (12.5) [16.2; 28.2]	0.406
	12.0 (36)	24.0 (24)	
	**CNV CSC, *N* (%)**	**Non-CNV CSC, *N* (%)**	
**FA**			***p* ^2^**
Hyperfluorescence leakage	3 (60.0%)	7 (36.8%)	0.616
Diffuse oozing	2 (40.0%)	7 (36.8%)	1.00
Both	0 (0.0%)	5 (26.3%)	0.544
**FIPED**			***p* ^2^**
Presence	4 (80.0%)	14 (73.3%)	0.399
Absence	1 (20.0%)	5 (26.3%)	0.455

FA: fluorescein angiography. FIPED: Flat irregular pigment epithelial detachment. Mean (Typical Dev.) [95% confidence interval of the mean] Median (Interquartile range) ^1^ Mann–Whitney Test ^2^ Chi-squared Test. * *p*-value < 0.05.

**Table 2 diagnostics-13-02792-t002:** BCVA at 12 months, 24 months, and 36 months after half-fluence photodynamic therapy in all affected eyes between groups: NVC-CSC and non-NVC CSC.

BCVA	CNV CSC (*N* = 5)	Non-CNV CSC (*N* = 19)	*p* ^1^
12-month visit	0.67 (0.22) [0.37; 0.96]	0.77 (0.27) [0.63–0.9]	0.367
	0.63 (0.45)	0.8 (0.37)	
24-month visit	0.63 (0.26) [0.37; 0.96]	0.80 (0.29) [0.66; 0.93]	0.297
	0.63 (0.45)	0.80 (0.20)	
36-month visit	0.56 (0.14) [0.37; 0.96]	0.75 (0.27) [0.61–0.87]	0.446
	0.63 (0.45)	0.8 (0.37)	
***p* ^2^**	0.368	0.174	

Best corrected visual acuity (BCVA). Mean (Typical Dev.) [95% confidence interval of the mean] Median (Interquartile range) ^1^ Mann–Whitney Test. ^2^ Friedman Test * *p*-value < 0.05.

**Table 3 diagnostics-13-02792-t003:** SS-OCT parameters at 12 months, 24 months, and 36 months after half-fluence photodynamic therapy in all affected eyes, in NVC-CSC and non-NVC CSC.

	CNV CSC (N = 5)	Non-CNV CSC (N = 19)	*p* ^1^
CFT			
12-month visit	219.5 (56.9) [128.9; 310.1]	303.4 (56.0) [277.2; 329.6]	0.018 *
	207.0 (107)	300.5 (104)	
24-month visit	219.5 (51.3) [137.8; 301.2]	297.2 (73.5) [262.7; 331.6]	0.046 *
	208.0 (96)	279.0 (122)	
36-month visit	230.6 (49.2) [169.6; 291.6]	297.8 (69.2) [264.4; 331.1]	0.043 *
	229.0 (97)	308.0 (128)	
***p* ^2^**	0.801	0.986	
**CRT**			
12-month visit	205.0 (12.1) [185.7; 224.3]	235.2 (74.0) [200.5; 269.8]	0.309
	206.5 (23)	217.0 (40)	
2-month visit	211.0 (15.4) [186.5; 235.5]	225.5 (56.6) [199.0; 252.0]	0.794
	218.0 (25)	217.0 (46)	
36-month visit	211.2 (18.1) [188.7; 233.7]	214.9 (44.4) [193.5; 236.4]	0.836
	217.0 (33)	218.0 (32)	
** *p* ** ** ^2^**	0.805	0.666	

Best corrected visual acuity (BCVA), central foveal thickness (CFT), and central retinal thickness (CRT). Mean (Typical Dev.) [95% confidence interval of the mean] Median (Interquartile range) ^1^ Mann–Whitney Test. ^2^ Friedman Test * *p*-value < 0.05.

## Data Availability

No new data were created or analyzed in this study. Data sharing is not applicable to this article.

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
