# Peer review of "Three-Year Follow-Up Detecting Choroidal Neovascularization with Swept Source Optical Coherence Tomography Angiography (SS-OCTA) after Successful Half-Fluence Photodynamic Therapy for Chronic Central Serous Chorioretinopathy"

_diagnostics, 2023, doi:10.3390/diagnostics13172792_

Round 1

Reviewer 1 Report

The authors reported the foveal image after PDT in the patients with chronic CSC. The study design is sound and the result presentation is clear. However, I think the Tables should be demonstrated in gray scale. Otherwiese, the paper is suitable for publication.

Minor editing of English language required

Author Response

Dear reviewer, I sincerely appreciate your feedback. I have implemented changes by converting the table colors to grayscale and making minor corrections to the English vocabulary. Your assistance is greatly valued.

Thank you in advance.

Reviewer 2 Report

The subject of the manuscript  is of interest to the reader ,  but the number of cases is so little to judge and make evidence based medicine and facts on CSR.

But the CSR is of interest to the reader as there is no approved  method of treatment except the yellow laser treatment .

5 out of 24 eyes developed CNV about 20% is a high incidence but the problem is the selection of old age confused the results.

The discussion is deep and comparing the other results with the author's results. 

Author Response

Dear reviewer, I extend my gratitude for your insightful comments, which I hold in high regard. As you pointed out, I have addressed the limitations section by acknowledging the limited number of cases and highlighting the importance of considering potential confounders, such as age, particularly given the high incidence of CNV in these cases.

Thank you in advance for your continued guidance.

Reviewer 3 Report

Chronic central serous chorioretinopathy is a common cause of decreased visual acuity in young men,
resulting from accumulation of subretinal fluid, atrophy of the outer layers of the retina or retinal pigment epithelium,
and flat pigment epithelial detachment mainly in the macular area. The paper describes the results after applying photodynamic therapy in 24 patients during 36 months of observation.
Visual acuity improved. The use of the SS OCT-A studies is a good documentation of the study. The results confirm the beneficial effect of the therapy on the condition of the retina.
The statistical analysis used does not raise any doubts. References cited correctly.

Author Response

Dear reviewer, I would like to express my appreciation for your valuable comments. Your insights are genuinely appreciated, and I am thankful for them. The results, as you have pointed out, are indeed intriguing.

Best regards.